# Improving Electromagnetic Interference Shielding While Retaining Mechanical Properties of Carbon Fiber-Based Composites by Introducing Carbon Nanofiber Sheet into Laminate Structure

**DOI:** 10.3390/polym14091658

**Published:** 2022-04-20

**Authors:** Yingjian Ma, Yangpeng Zhuang, Chunwei Li, Xing Shen, Liying Zhang

**Affiliations:** 1State Key Laboratory of Mechanics and Control of Mechanical Structures, Nanjing University of Aeronautics and Astronautics, Nanjing 210016, China; summeryuyi2008@163.com; 2Shanghai Collaborative Innovation Center of High Performance Fibers and Composite (Province-Ministry Joint), Center for Civil Aviation Composites, Donghua University, Shanghai 201620, China; 2200443@mail.dhu.edu.cn; 3AVIC General Huanan Aircraft Industry Co., Ltd., Zhuhai 519042, China; 18688170899@163.com

**Keywords:** electrospinning, carbon nanofibers (CNFs), electromagnetic interference (EMI), interlaminar shear strength (ILSS)

## Abstract

The demands for carbon fiber reinforced composites (CFRCs) are growing in the aviation industry for fuel consumption savings, despite the increasing risk of electromagnetic interference (EMI). In this work, polyacrylonitrile (PAN) sheets were prepared by electrospinning. Carbon nanofiber (CNF) sheets were obtained by the carbonization of PAN sheets. The laminate structures of the CF reinforced bismaleimide (BMI)-based composites were specially designed by introducing two thin CNF sheets in the upper and bottom plies, according to EMI shielding theory. The results showed that the introduction of CNF sheets led to a substantial increase in the EMI shielding effectiveness (SE) by 35.0% compared with CFRCs free of CNF sheets. The dominant EMI shielding mechanism was reflection. Noticeably, the introduction of CNF sheets did not impact the interlaminar shear strength (ILSS) of CFRCs, indicating that the strategy provided in this work was feasible for fabricating CFRCs with a high EMI shielding performance without sacrificing their mechanical properties. Therefore, the satisfactory EMI shielding and ILSS properties, coupled with a high service temperature, made BMI-based composites a promising candidate in some specific fields, such as high-speed aircrafts and missiles.

## 1. Introduction

With the rapid development of electronic technology, the high demand for electronic devices and facilities is leading to a steady increase in electromagnetic (EM) radiation emitted from various electronic products. Some sensitive equipment, such as electronic products used in aerospace and military systems, may be subject to electromagnetic incompatibility, causing perturbations in their performance [1,2]. Currently, various electromagnetic interference (EMI) shielding materials have been developed to protect electronic devices from unwanted EM radiation [3,4,5]. The demand for novel materials with a more efficient EMI shielding performance is growing.

Metallic materials were considered to be suitable EMI shielding materials at an early stage owing to their excellent electrical conductivity [6]. However, they are not widely used in practical applications because of their poor corrosion resistance, low strength-to-weight ratio and high cost [7]. Polymer composites with conductive fillers, such as metallic and carbon fibers, were thus developed [8,9]. Because of their attractive merits, including light weight, flexibility and easy processing, polymer composites have become promising candidates for the replacement of metallic materials [10,11,12]. Recently, carbon fiber (CF)-reinforced composites (CFRCs) have received considerable attention. Commercial aircraft manufacturers, such as Boeing, Airbus and COMIC, employed large amounts of CFRCs to replace conventional aluminum and titanium alloys in some primary structures because CFRCs possessed outstanding mechanical strength-to-weight ratios [13]. However, the EMI shielding property was sacrificed with the growth of the CFRCs used in aircraft. Because the electrical conductivity of CFRCs is lower than that of aluminum or titanium alloys, a satisfactory EMI shielding effectiveness (SE) cannot be attained [14]. Incorporating conductive fillers into CFRCs is a common approach for improving EMI SE. Hu et al. prepared carbonyl iron powder-CF felt/epoxy resin composites [15]. Although excellent EMI SE was obtained, a high content of carbonyl iron powder (37.5 wt%) was required, causing the resultant composites to suffer from poor flexibility, high density and high cost. Metal coating of CFs is another effective approach to improve the electrical conductivity of CFRCs. Kim et al. [16] and Zhu et al. [14] fabricated CFRCs containing nickel (Ni)-coated CFs, achieving SE values higher than 30 dB, which was adequate for practical applications. However, it has been reported that the metal coating approach produces some defects on CF [17] and weakens the interfacial adhesion between fibers and the matrix [18]. 

In this work, we aim to fabricate exceptional EMI shielding CFRCs without sacrificing their mechanical properties. Carbon nanofiber (CNF) sheets obtained from the carbonization of electrospun polyacrylonitrile (PAN) sheets served as a reinforcement in CFRCs. The laminating structure of CNF sheets and CF prepreg was specifically designed based on EMI shielding theory. The EMI shielding performance and interlaminar shear strength (ILSS) were investigated, and their corresponding mechanisms were discussed in detail.

## 2. Experiments

### 2.1. Materials

Polyacrylonitrile (PAN, M_W_ = 51,000) and *N*,*N*-dimethylformamide (DMF) solutions were supplied by Rhawn Chemical Technology Co., Ltd., Shanghai, China. Unidirectional prepreg (CCF800H/AC631) composed of CF and bismaleimide (BMI) was provided by AVIC Composite Co., Ltd., Beijing, China. The density of the CF is 133 ± 5 g/m^2^, and the resin content of prepreg is 33 ± 2 wt%. The nominal ply thickness of the prepreg is 0.125 mm.

### 2.2. Preparation of Carbon Nanofiber Sheets

10 wt% PAN precursor solution was prepared by dissolving certain amounts of PAN in DMF with magnetic stirring at 1000 rpm for 8 h. The PAN nanofiber sheets were prepared from the precursor solution using electrospinning technology in a conditioned room at 25 °C and 39% relative humidity (RH). The feeding rate was 0.4 mL/h, and the voltage was set at 15 kV. The tip-to-collector distance was 15 cm, and the speed of the rotating mandrel was 400 rpm. Afterwards, the PAN nanofibrous sheets were placed between two graphite plates for carbonization. The sheets were subjected to pre-oxidation treatment at 280 °C and welted for 1 h under an air atmosphere in a tube furnace. Subsequently, the sheets were subjected to a heat treatment at 1000 °C and welted for 1 h under continuous argon purging. After the tube furnace was cooled to room temperature, the carbon nanofiber (CNF) sheets were obtained.

### 2.3. Preparation of Composites

First, the prepreg was stacked with a layup sequence to form a preform, which is composed of 8 layers of 0.125 mm-thick 0° prepreg. The preform was named P in this work. Second, two different samples (i.e., P-P and C-P-P-C) were prepared according to the layups illustrated in Figure 1. Here, a CNF sheet was labeled *C*. Finally, the composites were prepared by a vacuum bag molding process. The preforms were heated to 130 °C at a rate of 5 °C/min and held for 1 h with a vacuum pressure of 0.098 MPa in an oven. Afterwards, the temperature was heated to 180 °C at the same heating rate and held for 2 h. The temperature was then raised to 200 °C for post-curing for 6 hours. Finally, the composites were removed from the vacuum bag after the mold was cooled to room temperature.

### 2.4. Characterizations

The morphologies of PAN sheets, CNF sheets, and composites were observed using scanning electron microscopy (SEM, SU-4800). The crystal structures of PAN nanofibers and carbon nanofibers (CNFs) were analyzed by X-ray diffraction (XRD, Bruker D8). Raman microscopy (Raman, LabRAM HR Evolution) was employed to analyze the structure of PAN nanofibers and CNFs. The electrical conductivities of the sheets and composites were measured using an avometer (DMM6500, KEITHLEY) according to ASTM D4496 and ASTM D257, respectively. The EMI shielding measurements were carried out in the frequency range of the X-band (8.2–12.4 GHz) by using a vector network analyzer (VNA, ZNB 20). The samples were fabricated into rectangle plates of 22.86 mm × 10.16 mm × 2.2 mm to fit the WR 90 waveguide. The interlaminar shear strength (ILSS) of the composites was measured using a universal testing machine (ETM 105D, WANCE) according to ASTM D2344. The dimensions of the composites were 20 mm × 6 mm × 2 mm, while the gauge length was 8 mm. All the tests were recorded at least five times.

## 3. Results and Discussion

Figure 2 shows the representative SEM micrographs of PAN nanofibers and CNFs. The smooth and straight PAN nanofibers were attained with an average diameter of 129 ± 18 nm. After carbonization, unevenly distributed CNFs with thinner diameters (68 ± 16 nm) were obtained. In addition, as shown in Figure 2a, random PAN nanofibers were toughed together. During carbonization, a series of complex reactions, such as cyclization, pyrolysis and condensation, occurred, leading to the rearrangement of the molecular structures [19]. As a result, the areas where the PAN nanofibers were tough tended to merge together, forming a continuous two-dimensional network (Figure 2b), which is beneficial to the improvement of electrical conductivity.

Figure 3a shows the XRD spectra of PAN and CNF sheets. Two diffraction peaks were located at 17.0° and 29.0°, which can be assigned to the (200) and (020) crystal planes of PAN [20]. After carbonization, the peak that appeared at 26.5° was attributed to the (002) diffraction, confirming the formation of the turbostratic and polyaromatic structure of CNF [21]. Figure 3b shows the Raman curves of PAN and CNF sheets. The bands of PAN nanofiber that appeared at 1320, 1355, and 1450 cm^−1^ were ascribed to the vibrations of methylene groups and methine groups [22]. A sharp peak at 2244 cm^−1^ corresponding to the typical nitrile group of PAN was also observed [23]. The nitrile group disappeared after carbonization and was replaced by two typical peaks at 1360 and 1590 cm^−1^, which were attributed to the stretching of the sp^2^ hybridized carbon atoms and the structural defects and disorder in the hexagonal structure, respectively. This observation indicated that CNF sheets were successfully obtained.

According to EMI shielding theory, the interaction of more mobile charge carriers of materials leads to the strong reflection of EM waves [24]. Thus, the composites were fabricated with a special stacking sequence, where the upper and bottom plies were CNF sheets and the prepreg was in the middle, as shown in Figure 4. The thicknesses of the upper and bottom CNF sheets were almost the same and were measured to be ~15 μm. From the enlarged SEM image, the resin was fused into the CNF sheets during the preparation process. The total EMI SE can be calculated using the following equation [25]:(1)$$SE_{total}=-10lgT=-10lg{\left|S_{21}\right|}^{2}=-10lg{\left|S_{12}\right|}^{2}$$
where *T* is the power coefficient of transmission, and *S*_21_ and *S*_12_ are the forward and reverse transmission parameters, respectively. The EMI SE in the X band and electrical conductivities of P-P and C-P-P-C are presented in Figure 5a. The electrical conductivity and *SE* value demonstrated a strong correlation. The average SE values of P-P and C-P-P-C were 17.7 and 23.9 dB, respectively. The 35.0% improvement was mainly attributed to the incorporation of CNF sheets.

The EMI shielding mechanisms were studied to examine the effects of CNF sheets. The total EMI SE (*SE_total_*) is composed of reflection (*SE_R_*), absorption (*SE_A_*) and multiple reflection (*SE_MR_*), which can be expressed as:(2)$$SE_{total}=SE_{A}+SE_{R}+SE_{MR}$$
in which *SE_MR_* can be negligible when *SE_total_* ≥ 15 dB [26]. In this work, the *SE_total_* of P-P and C-P-P-C was higher than 15 dB; thus, Equation (2) can be simplified as:(3)$$SE_{total}\approx SE_{A}+SE_{R}$$

*SE_R_* and *SE_A_* can be derived from the following equations:(4)$$SE_{R}=10\times lg\left(1/\left(1-R\right)\right)$$
(5)$$SE_{A}=10\times lg\left(\left(1-R\right)/T\right)$$
(6)$$R={\left|S_{11}\right|}^{2}={\left|S_{22}\right|}^{2}$$
(7)A=1−R−T
where *R* and *A* are the power coefficients of reflection and absorption, respectively. According to the above equations, the *SE_R_* and *SE_A_* of the composites are plotted in Figure 5b. Although *SE_A_* was higher than *SE_R_* for both P-P and C-P-P-C, it is improper to determine that absorption was the primary EMI shielding mechanism because *SE_A_* refers to the capability of the material to attenuate the incident power that has entered the materials [27]. Power coefficients were adopted to compare the contribution of reflection and absorption to the shielding performance. It can also be observed from Figure 5b that *R* ≫ *A* for both P-P and C-P-P-C, indicating that the amount of incident power was mainly blocked by reflection. Additionally, the values of *SE_total_*, *SE_R_*, *SE_A_*, *R* and *A* were almost the same due to the symmetric structure of the C-P-P-C. These results were consistent with our assumption. Figure 5c illustrates the EMI shielding mechanisms of C-P-P-C. When an incident EM wave struck the first ply of the CNF sheets, a large amount of EM wave was reflected because the CNFs sheet with a 2D conductive network enhanced the mobility and amounts of free electrons, which facilitated the interaction between EM waves and the electrons. The rest of the EM wave penetrated the CNFs sheet and was multiply reflected and scattered among CFs, ultimately depleting the EM energy by converting into thermal energy inside the composites. Only small amounts of EM waves arrived at the bottom ply of the CNF sheets and were reflected in the composites. The reflected EM wave was attenuated among CFs as well, leading to the improvement of EMI SE. As a result, it can be concluded that reflection was the dominant EMI shielding mechanism of C-P-P-C.

To assess the real applications of the composites, the ILSS, which is one of the crucial parameters for qualifying the mechanical properties of CFRCs, was tested. Figure 6a shows that the ILSSs of P-P and C-P-P-C were 88.3 and 87.0 MPa, respectively. The slight decrease might be attributed to the increase in porosity generated during the fabrication of C-P-P-C. The dominant failure mechanism of both P-P and C-P-P-C was interlaminar delamination, as shown in Figure 6b,c. These results indicated that the role of CNF sheets could be ignored in determining the mechanical properties of CFRCs.

Table 1 shows the comparison of the BMI-based CF reinforced composites with other recently reported thermoset resin-based CF reinforced composites. Typically, the EMI shielding performance was compared through the SE total normalized by the sample thickness (*d*) [28]. It is clear that some epoxy-based composites exhibited a slightly higher value of *SE_total_*/*d* [15,29,30,31], but the ILSS property was not characterized. Compared with the work done by Zhu et al. [14], the value of *SE_total_*/*d* was higher than that of BMI-based composites. However, the ILSS property was inferior. Furthermore, the service temperature of BMI-based composites was much higher than that of epoxy-based composites. Thus, we believe that our composites can be a promising candidate in some specific areas.

## 4. Conclusions

In this work, we developed a structural design strategy for improving EMI SE while retaining the mechanical properties of CFRCs. CNF nanofiber sheets were prepared by the carbonization of PAN nanofiber sheets and then incorporated into the upper and bottom plies of the composites according to EMI shielding theory. The CFRCs with special lamination sequences exhibited an EMI SE of 23.9 dB, which was improved by 35.0% compared with their counterparts. Most of the incident EM waves interacted with CNF nanofiber sheets and were attenuated by reflection. Noticeably, the ILSS remained almost unchanged after incorporating CNF nanofiber sheets. Because the resultant CFRCs exhibited both excellent EMI SE and mechanical properties, it is believed that this would be a feasible strategy for guiding the structural design of composites.

## Figures and Tables

**Figure 1 polymers-14-01658-f001:**
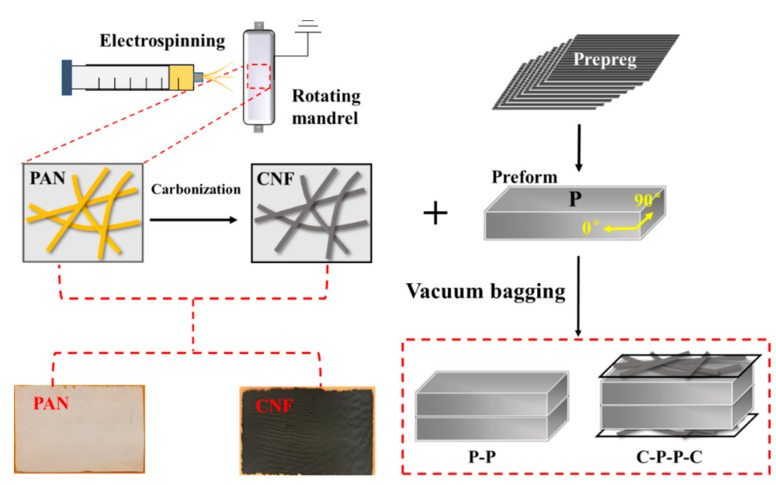
Schematic illustration of the preparation process of composites.

**Figure 2 polymers-14-01658-f002:**
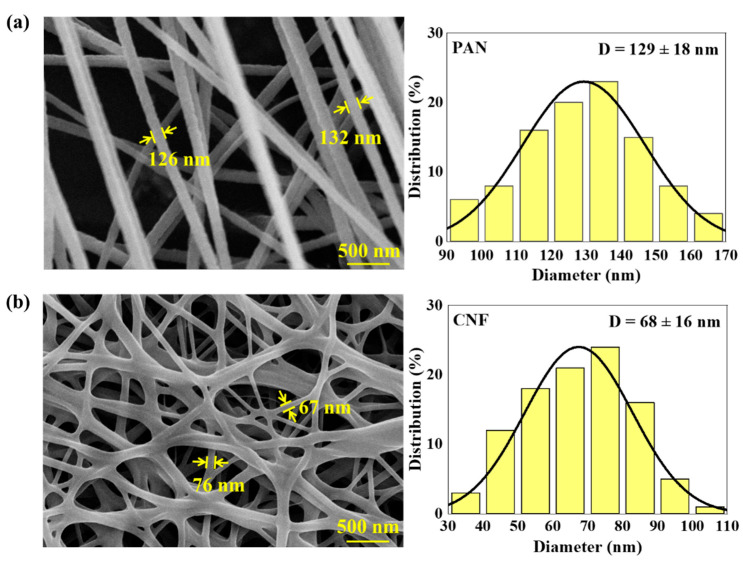
Representative SEM images of electrospun nanofibers (**a**) before and (**b**) after carbonization.

**Figure 3 polymers-14-01658-f003:**
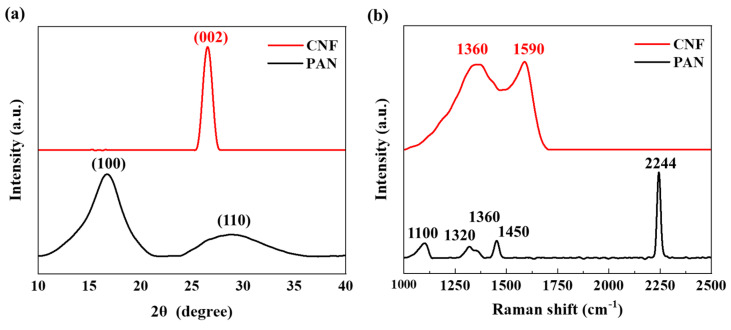
(**a**) XRD spectra and (**b**) Raman curves of PAN nanofibers and CNFs.

**Figure 4 polymers-14-01658-f004:**
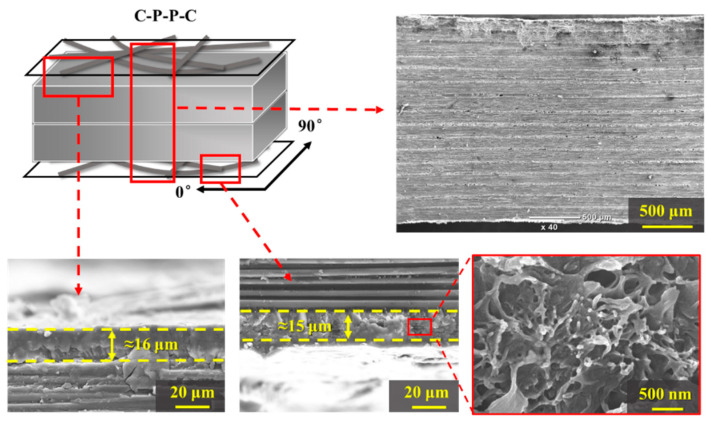
Microstructure of the composites with a special stacking sequence.

**Figure 5 polymers-14-01658-f005:**
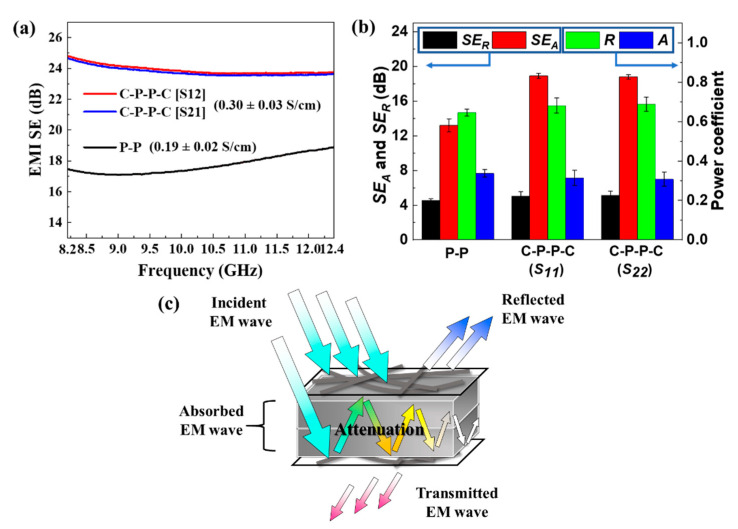
(**a**) EMI SE and electrical conductivities of P-P and C-P-P-C calculated using *S*_12_ and *S*_21_; (**b**) *SE_A_*, *SE_R_* and power coefficient of P-P and C-P-P-C calculated using *S*_11_ and *S*_22_; (**c**) schematic illustration of EMI shielding mechanisms of C-P-P-C.

**Figure 6 polymers-14-01658-f006:**
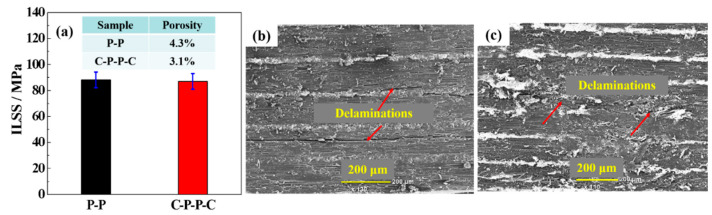
(**a**) ILSS of P-P and C-P-P-C; SEM images of fracture surfaces of (**b**) P-P and (**c**) C-P-P-C.

**Table 1 polymers-14-01658-t001:** Comparison of EMI SE and ILSS of the recently reported CF reinforced composites.

Sample	Functional Fillers (Content)	Thickness (mm)	EMI SE (dB)	ILSS (MPa)	Ref.
CF/Epoxy	Nylon-66 nanofiber (0.25 wt%)	4.4	62.7	/	[29]
CF/Epoxy	Carbonyl iron powders (/)	4.0	53.9	/	[15]
CF/Epoxy	Ni-PDA coating (15.4 wt%)	1.0	31	61.2	[14]
CF/Epoxy	TAPc NWs@GO (0.5 wt%)	2.0	26	/	[30]
CF/Epoxy	B-MWNTs (1.0 wt%)	0.5	65	/	[31]
CF/Epoxy	Fe_3_O_4_@GO (/)	/	46.3	70.9	[32]
CF/BMI	CNFs (0.37 wt%)	2.2	23.9	88.3	Our work

(Ni-PDA: nickel (Ni)-attached polydopamine (PDA); TAPc NWs@GO: tetraamino-phthalocyanine nanowires (TAPc NWs) decorated on graphene oxide (GO); B-MWNTs: branched poly(ethyleneimine) functionalized multiwalled carbon nanotubes; Fe_3_O_4_@GO: Fe_3_O_4_-deposited GO).

## Data Availability

The data used to support the findings of this study are available from the corresponding authors upon request.

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
