# Peer review of "Improving Electromagnetic Interference Shielding While Retaining Mechanical Properties of Carbon Fiber-Based Composites by Introducing Carbon Nanofiber Sheet into Laminate Structure"

_polymers, 2022, doi:10.3390/polym14091658_

Round 1
Reviewer 1 Report
This gives a new material for electromagnetic shielding and as such it has a useful result. Some points to consider are 1) All abbreviations need to be spelled out (some missing seem to be PANF, PANM, ILSS), 2) It would be most useful to show |S11| and |S21], 3) Due to symmetry S12=S21 and S11=S22, but it would seem almost as effective if the duplicate bottom layers were omitted as long as the top layer were the one exposed to the incoming signal, 5) The flexibility would be of interest, for example in wrapping around a body part; maybe a comment on flexibility can be given.
Author Response
We appreciate the reviewers in reviewing this manuscript. We have carefully studied all the comments. Below is our point-by-point response to address the questions raised. The necessary changes were made and highlighted in the revised manuscript accordingly. For those parts remain unchanged, we give our explanations in the replies. We trust the revised version is ready for publication.

Reviewer 2 Report
The manuscript "Improving electromagnetic interference shielding while retaining mechanical properties of carbon fiber-based composites by introducing carbon nanofiber sheet into laminate structure" shows an interesting subject to obtain EMI shieldings by including CNF in PAN in laminates.
There are parts that needs to be revised especially the introduction lacks of references.
Page 1, line 35,. Please add reference after radiation, line 38 add reference after conductivity, line 41 add reference after developed.
page 2 line 86. What means [0]8. Its confusing please explain it in better form
Page 4 line 127. The sharp peak in the Raman spectra is not 2240, its 2244 as shown in spectra please correct such, or reformulate
Figure 5b. please add standard deviations
it would be beneficial adding a table of the EMI shieldings contained in this work in comparison to other to compare how well those achieved in this work. Please add those before conclusions
Author Response

(The authors gave the same response as above.)

Reviewer 3 Report
Authors reported the synthesis of carbon nanofiber sheets by carbonization of PAN sheets, and subsequent introduced two thin CNF sheets in the upper and bottom plies to form laminate structures. The EMI performance showed that a substantial increase in EMI shielding effectiveness by 35.0% compared with CFRCs free of CNF sheets. The performace sounds good and a series of results are also discussed. However, some issues should be addressed.
1, In Abstract section, the significance and futural application of the present materials were not mentioned. In Experimental section, the molecular weight of PAN should be descriped after the product introduction.
2, In SEM section, it is mentioned that “it can also be seen that CNFs were fused together to form a continuous two-dimensional network...”. So, what caused the formation of a continuous two-dimensional network? The deep mechanisms should be investigated since it is of importance for enhanced conductivety.
3, Some key and important research results in EM absorption should be mentioned and cited so that we can provide a solid background and progress to the readers, such as Journal of Materials Chemistry C, 2016, 4, 9738; ACS Applied Materials & Interfaces, 2017, 9, 16404; Composites Part A, 2018, 115, 371.
4, In the present work, it is unable to provide a satisfactory modelling of the EMI mechanism (s): To what is due EMI (conductivety, structure, absorption, reflection, scaterring...)? This fundamental issue is not all answered. Please investigate the EMI properties from the mechanism views.
5, In case of real-life application, the ILSS was studied to illustrate the main failure mechanism of samples. And These results indicated that the role of CNF sheets could be ignored in determining the mechanical properties of CFRCs. As far as I am concerned, it is just to meet the application request, and how to achieve the strength requirements in practical application, please give more details.
Author Response

(The authors gave the same response as above.)
